# 6-Methoxymellein Isolated from Carrot (*Daucus carota* L.) Targets Breast Cancer Stem Cells by Regulating NF-κB Signaling

**DOI:** 10.3390/molecules25194374

**Published:** 2020-09-23

**Authors:** Ren Liu, Hack Sun Choi, Su-Lim Kim, Ji-Hyang Kim, Bong-Sik Yun, Dong-Sun Lee

**Affiliations:** 1Interdisciplinary Graduate Program in Advanced Convergence Technology and Science, Jeju National University, Jeju 63243, Korea; liuren0308@gmail.com (R.L.); ksl1101@naver.com (S.-L.K.); seogwi12@naver.com (J.-H.K.); 2Subtropical/Tropical Organism Gene Bank, Jeju National University, Jeju 63243, Korea; choix074@jejunu.ac.kr; 3Division of Biotechnology, College of Environmental and Bioresource Sciences, Jeonbuk National University, Gobong-ro 79, Iksan 54596, Korea; bsyun@jbnu.ac.kr; 4Practical Translational Research Center, Jeju National University, Jeju 63243, Korea; 5Faculty of Biotechnology, College of Applied Life Sciences, Jeju National University, SARI, Jeju 63243, Korea

**Keywords:** breast cancer stem cells (BCSCs), 6-methoxymellein, NF-κB p65, NF-κB p50, IL-6, IL-8

## Abstract

The presence of breast cancer stem cells (BCSCs) induces the aggressive progression and recurrence of breast cancer. These cells are drug resistant, have the capacity to self-renew and differentiate and are involved in recurrence and metastasis, suggesting that targeting BCSCs may improve treatment efficacy. In this report, methanol extracts of carrot root were purified by means of silica gel, Sephadex LH-20, and preparative high-performance liquid chromatography to isolate a compound targeting mammosphere formation. We isolated the compound 6-methoxymellein, which inhibits the proliferation and migration of breast cancer cells, reduces mammosphere growth, decreases the proportion of CD44^+^/CD24^−^ cells in breast cancer cells and decreases the expression of stemness-associated proteins c-Myc, Sox-2 and Oct4. 6-Methoxymellein reduces the nuclear localization of nuclear factor-κB (NF-κB) subunit p65 and p50. Subsequently, 6-methoxymellein decreases the mRNA transcription and secretion of IL-6 and IL-8. Our data suggest that 6-methoxymellein may be an anticancer agent that inhibits BCSCs via NF-κB/IL-6 and IL-8 regulation.

## 1. Introduction

Carrot (*Daucus carota* L.) is one of the most important root vegetables cultivated worldwide and has health benefits [1,2,3,4]. With increasing health problems, carrots are becoming more popular plants because of their healthy nutrients and benefits for human health [5]. Phytochemicals, which comprise carotenoids, ascorbic acid, polyacetylenes and phenolic compounds, such as β-carotene, lutein, l-ascorbic acid, falcarinol and caffeic acid, contribute to the dietary value of carrots [4]. These compounds have also been shown to be potent inhibitors of inflammation and oxidative stress [6,7,8,9,10,11,12].

Breast cancer is a cancer mostly detected in females and is one of the major causes of female death [13]. It has been increasingly recognized that breast cancer is a malignancy displaying frequent inter- and intratumor heterogeneity [14]. For this reason, histological stratification is used to classify breast cancer based on progesterone receptor, estrogen receptor, and erbB-2 receptor (HER2) expression. Chemotherapy, hormone therapy, immunotherapy, radiotherapy, and surgery are the common modalities for breast cancer [15]. However, breast cancer stem cells (BCSCs) endowed with self-renewal capacity have been demonstrated to contribute to tumor heterogeneity. Multiple independent studies have shown the presence of distinct CSC populations within tumors based on the expression of CSC markers such as CD44^+^/CD24^−^ and aldehyde dehydrogenase (ALDH) [16,17]. Compared with non-CSCs, CSCs exhibit overactivation of transcription factors and proteins related to signal pathways, such as stemness markers: Sox-2, Oct4, c-Myc; Hedgehog and Wnt pathways [18,19,20,21,22]. c-Myc is important for regulating proliferation and the survival of glioma cancer stem cell [23]. Oct4 is responsible for breast CSC specification and regulated Nanog, Sox2 and Klf4 gene [24]. Sox2 drives cancer stemness and fuels tumor initiation [25]. Many studies show that the tumor microenvironment plays an important role in regulating CSC formation and tumor progression [26,27].

BCSC formation can be regulated by cytokines and cell types present in the tumor microenvironment, including mesenchymal stem cells, cancer-associated fibroblasts and tumor-infiltrating lymphocytes [28]. Additionally, the tumor microenvironment contains several noncellular components, including cytokines and growth factors. In particular, cytokines contribute to chronic inflammation, which promotes cancer cell survival and cancer disease progression by suppressing immune cell functions [29,30,31]. For instance, interleukin-6 (IL-6) is sufficient for converting non-CSCs to CSCs in different breast cells [30], and IL-6 regulates the conversion of non-CSCs into CSCs [31], activating the Notch-3-dependent upregulation of the Notch ligand Jagged-1 in breast cancer cells [32]. Interleukin-8 (IL-8) induces BCSC activity and chemoresistance in triple-negative breast cancer (TNBC) cells [33]. The IL-8 signaling pathway is mediated via an EGFR/HER2-dependent pathway [34]. Nuclear factor-κB (NF-κB) subunit p65, a regulator of IL-6 and IL-8, was suggested as a potential target against BCSCs [35,36].

Herein, we chose a colorful food, *D. carota* L., for isolating mammosphere inhibitors against BCSC. We isolated a compound, 6-methoxymellein that inhibits the mammosphere formation of breast cancer cell lines. We demonstrate that 6-methoxymellein suppresses BCSC formation through the NF-κB signaling pathway.

## 2. Results

### 2.1. Isolation and Identification of a Breast CSC Inhibitor Derived from D. carota L.

Bioassay-guided fractionation was performed to screen and isolate BCSC inhibitor from *Daucus carota* L extracts using a mammosphere formation assay from breast cancer cell lines. The methanol extracts of *D. carota L.* were purified via ethyl acetate extraction (*v*/*v* = 1:1), silicon dioxide gel, gel filtration chromatography, and preparatory HPLC (Figure 1A). CSC formation was suppressed by the purified compound (Figure 1B). The purity of the compound was confirmed using HPLC (Figure 1C). NMR and GC-MS data identified the compound as 6-methoxymellein (Figure 2).

### 2.2. 6-Methoxymellein Suppresses the Growth of Breast Cancer Cells and Mammospheres

The growth inhibitory effect of 6-methoxymellein was examined using increasing concentrations in breast cancer cell lines. Treatment with 6-methoxymellein for 24 h induced suppression of proliferation at >0.8 mM (MDA-MB-231) and >0.5 mM (MCF-7) (Figure 3A,B). To evaluate whether 6-methoxymellein inhibits mammosphere formation, primary mammospheres were treated with 6-methoxymellein. Compared to the control, 6-methoxymellein decreased the sphere number and decreased the size of the mammospheres (Figure 3C,D). In addition, 6-methoxymellein inhibited the formation of colony and cell migration (Figure 3E,F). We show that 6-methoxymellein suppresses cell migration, growth, colony formation, and mammosphere formation.

### 2.3. 6-Methoxymellein Reduces the Proportion of CD44^+^/CD24^−^-Expressing Breast Cancer Cells

The cell surface marker of breast CSCs is CD44^+^/CD24^−^ of breast cancers. The CD44^+^/CD24^-^ subpopulation of cancer cells was determined under 6-methoxymellein. 6-Methoxymellein decreased the proportion of CD44^+^/CD24^−^ MDA-MB-231 cancer cells from 80.3% to 41.6% (Figure 4).

### 2.4. 6-Methoxymellein Inhibits the Protein Expression of Cancer Stem Cell-Specific Markers and Inhibits Mammosphere Growth

Next, we analyzed whether 6-methoxymellein reduces the protein level of CSC marker genes. Indeed, 6-methoxymellein decreased the protein expression levels of cancer stem cell marker genes (Figure 5A). To examine whether 6-methoxymellein reduced mammosphere growth, mammospheres were cultured with 6-methoxymellein, and the number of cancer cells of mammospheres was examined. The results indicated that 6-methoxymellein inhibited mammosphere growth (Figure 5B).

### 2.5. 6-Methoxymellein Suppresses the Nuclear Localization of NF-κB p65 and NF-κB p50 in BCSCs

To determine the cellular mechanism by which 6-methoxymellein reduces mammosphere formation, we examined the total and nuclear protein levels of NF-κB p65 and NF-κB p50. We showed that the nuclear level of NF-κB p65 and NF-κB p50 were significantly reduced after 6-methoxymellein treatment (Figure 6A,B). Furthermore, we tested the direct binding of the NF-κB DNA probe to nuclear NF-κB proteins by electrophoretic mobility shift assay (EMSA) under 6-methoxymellein treatment (Figure 6C). We indicated that the amount of NF-κB protein bound to the NF-κB probe (arrow) was reduced under 6-methoxymellein treatment (Figure 6C, #3). The specificity of the NF-κB/probe complex was analyzed by using a 100X self-competitor oligo (Figure 6C, #4) and a 100X mutated NF-κB competitor (Figure 6C, #5).

### 2.6. 6-Methoxymellein Decreases the mRNA and Protein Levels of Secretory IL-6 and IL-8 in Mammospheres

It was investigated whether the activity of NF-κB is associated with IL-6 and IL-8 secretion [37,38,39]. To examine whether 6-methoxymellein decreases the level of IL-6 and IL-8 cytokine secretion, we determined the concentrations of secreted IL-6 and IL-8 in mammosphere culture medium using a human inflammatory assay. 6-Methoxymellein reduced the concentrations of the cytokines IL-6 and IL-8 (Figure 7A). Subsequently, we checked the levels of IL-6 and IL-8 transcript under 6-methoxymellein, and our data showed that 6-methoxymellein decreased the mRNA expression of IL-6 and IL-8 (Figure 7B).

## 3. Discussion

Carrot (*D. carota L.*) is the most important root vegetable, is cultivated worldwide, and is rich in natural phytochemicals. Carrot root is widely utilized because of abundant carotenoids, anthocyanins and dietary fiber. Carrots are the main source of carotenoids, and previous studies have demonstrated that carrots may reduce cancer risk and play an important role in a cancer prevention diet [40,41,42]. Carrots are traditionally used for the treatment of gastric ulcers, diabetes, muscle pain and cancer in Lebanon [43]. Carrots have been reported to provide numerous biological activities, including antibacterial, antifungal, diuretic, antilithic, anticancer, antiinflammatory and antioxidant effects [43,44,45,46,47]. In this study, we purified a BCSC inhibitor from carrots (Figure 1) and identified as 6-methoxymellein by electrospray ionization (ESI)-MS and NMR spectroscopy (Figure 2). 6-Methoxymellein belongs to the mellein family, which is a subgroup of 3,4-dihydroisocoumarins. Previously, mellein was produced by the fungus *Aspergillus ochraceus*. Coumarin and isocoumarin are products of bacteria, fungi, plants, insects, lichens, and marine sponges. These compounds have different biological functions, including antimicrobial, anticancer, antileukemia and antivirus activities [48,49,50]. 6-Methoxymellein was the first mellein derivative isolated from carrots in 1960 [51]. Subsequently, it was purified from carrot root cultured with *Ceratocystis cimbriata*, *Helminthosporum carbonum*, and *Fusarium oxysporum* [52]. It was supposed that 6-methoxymellein production resulted from a change in plant metabolism induced by the fungi [53], suggesting that 6-methoxymellein induces the active defenses of carrot-based compounds against fungi; thus, 6-methoxymellein was classified as a phytoalexin [53,54].

Targeting cancer stem cells that contribute to therapy resistance, metastasis, and recurrence is a challenge of breast cancer treatment [55,56]. Mammosphere formation assays were performed to identify functional CSCs in vitro [57]. Our results show that 6-methoxymellein suppressed the growth of breast cancer cells, decreased the size and efficiency of mammospheres formation (Figure 3C,D), and inhibited cell migration and colony formation (Figure 3E,F). In addition, 6-methoxymellein reduced the CD44^high^/CD24^−^ subpopulation in MDA-MB-231 cells (Figure 4). To address the effect of 6-methoxymellein on the stemness of mammospheres, we checked the protein levels of c-Myc, Oct4 and Sox-2 in mammospheres under 6-methoxymellein treatment. Our data show that 6-methoxymellein inhibits c-Myc, Oct4 and Sox-2 expression and reduces mammosphere growth (Figure 5A,B). It has been reported that the regulatory mechanisms of BCSC formation are extensive and complex. Several pathways, such as the Notch, Wnt, NF-κB, JAK/STAT and Hedgehog pathways, are involved in the maintenance of stemness [18]. In addition, the tumor *microenvironment plays an essential role in supporting and maintaining CSCs* [28,58,59]. Recently, cytokines were reported to regulate the self-renewal and survival of breast CSCs in the tumor microenvironment [60]. MSCs interact with BCSCs through IL-6 and chemokine (C-X-C motif) ligand 7 cytokine secretion. This signaling is responsible for the self-renewal potential of BCSCs. Subsequently, CXCL7 secreted by MSCs promotes cancer stem cell resistance to anticancer drugs [61]. IL-8 regulates breast cancer stem cell activity by binding to C-X-C motif chemokine receptor 1/2. Targeting CXCR1/2 proteins reduces breast CSC activity and increases their ability to inhibit HER2 [34]. The NF-κB signaling pathway plays a role in inflammation and tumorigenesis.

The secretion of IL-6 and IL-8 can be regulated by the NF-κB signaling pathway [62]. 6-Methoxymellein inhibited the localization of NF-κB p65 and NF-κB p50 in the nucleus but did not affect the total level of NF-κB p65 and NF-κB p50 proteins (Figure 6A,B). In addition, the nuclear NF-κB DNA binding ability was inhibited by 6-methoxymellein (Figure 6C). Subsequently, 6-methoxymellein reduced the mRNA transcription and secretion of IL-6 and IL-8 (Figure 7). These data suggest that 6-methoxymellein may be used as an anticancer agent and exerts its effects through the NF-κB and cytokine signaling pathways.

## 4. Materials and Methods

### 4.1. Chemicals

The silica powder and plate were obtained from Merck (Darmstadt, Germany). The Sephadex LH-20 resin was purchased from Sigma-Aldrich (St. Louis, MO, USA). The EZ-Cytox cell viability kit was obtained from DoGenBio (Seoul, Korea). The other chemicals were purchased from Sigma-Aldrich (St. Louis, MO, USA).

### 4.2. Plant Material

The carrot sample was obtained from farmers (Seogwipo, Jeju, Korea), washed with tap water, dried, and ground. The carrot sample (No. 2019_03) has been deposited at the Department of Biotechnology, Jeju National University (Jeju, Jeju-si, Korea).

### 4.3. Isolation of Mammosphere Formation Inhibitor from Carrots

We followed a previously described method [35]. The ground carrot sample (1.2 kg) was suspended and extracted with methanol. The purification protocol is described in Figure 1. The carrot powder (1.2 kg) was extracted with 12 L of methanol. The methanol extracts were concentrated to 2 L and saturated with H_2_O, and the methanol part was evaporated. The water part was extracted with an equal volume of ethyl acetate (*v*/*v* = 1:1). The ethyl acetate fraction was recovered and loaded on a silica gel column (3 × 35 cm), and the sample was fractionated with solvent (chloroform-methanol, 10:1) (Appendix A). The five fractions were recovered and tested by mammosphere assay. The 1 and 2 fractions suppressed mammosphere formation. Parts 1 and 2 were recovered and loaded on a Sephadex LH-20 column (2.5 × 30 cm) and fractionated into three parts (Appendix A). The three fractions were recovered and tested by mammosphere assay. Fraction 3 suppressed mammosphere formation. Fraction 3 was fractionated using preparatory TLC (glass plate; 20 × 20 cm) and developed in a TLC glass chamber (chloroform-methanol, 100:1). The main band was isolated from the silica gel plate, and the fraction was tested by mammosphere formation assay (Appendix A). The major fraction was analyzed using a Shimadzu HPLC LC-20 (Shimadzu, Tokyo, Japan). HPLC was performed using a C18 column (10 × 250 mm, flow rate; 2 mL/min). For isolation, the acetonitrile concentration was initially set at 0%, increased to 60% at 15 min and finally increased to 100% at 40 min (Appendix A).

### 4.4. Structure Analysis of the Isolated Component

The molecular structure of the purified compound was analyzed by NMR and mass spectrometry. The molecular weight of the compound was determined to be 208 by ESI-mass spectrometry, which indicated a quasimolecular ion peak at m/z 209.3 [M + H]^+^ in positive mode (Appendix A). The ^1^H-NMR spectrum in CD_3_OD showed signals due to two aromatic methines at δ 6.35, one oxygenated methine at δ 4.66, one methoxy methyl at δ 3.82, a nonequivalent methylene at δ 2.96 and 2.84, and one methyl at δ 1.46. In the ^13^C-NMR spectrum, 11 carbon peaks included one carbonyl carbon at δ 171.7, two oxygenated sp^2^ quaternary carbons at δ 167.7 and 165.7, two sp^2^ methine carbons at δ 107.1 and 100.6, two sp^2^ quaternary carbons at δ 143.4 and 102.6, one oxygenated methine carbon at δ 77.4, one methoxy carbon at δ 56.3, one methylene carbon at δ 35.6, and one methyl carbon at δ 21.0 (Appendix A). All proton-bearing carbons were assigned by the heteronuclear multiple-quantum coherence (HMQC) spectrum, and the ^1^H-^1^H COSY spectrum showed one partial structure (Appendix A). Further structural elucidation was performed with the aid of the heteronuclear multiple bond correlation (HMBC) spectrum, which showed long-range correlations from the methine protons at δ 6.35 to the carbon at δ 102.6 and from the methylene protons at δ 2.94/2.84 to the carbons at δ 143.4, 107.1, and 102.6. Finally, the methoxy group was connected by the long-range correlation from the methyl protons at δ 3.82 to the carbon at δ 167.7 (Appendix A). The molecular structure of the purified compound was determined to be 6-methoxymellein (Figure 2).

### 4.5. Cell Line and Culture of Mammospheres

MCF-7 and MDA-MB-231 human breast cancer cells were purchased from the Korea Cell Line Bank (KCLB, Seoul, Korea) and incubated in Dulbecco’s modified Eagle’s medium (DMEM) supplemented with 10% fetal bovine serum (FBS) and 1% penicillin/streptomycin (Gibco, CA, USA) in an incubator (37 °C, 5% CO_2_). Human breast cancer cells were cultured at 0.5 × 10^4^ (MDA-MA-231) and 4 × 10^4^ (MCF-7) cells per plate in an ultralow attachment 6-well plate with a MammoCult^TM^ culture medium (StemCell Technologies, Vancouver, BC, Canada) supplemented with hydrocortisone and heparin for 1 week in an incubator (37 °C, 5% CO_2_). Mammosphere formation was assessed using the National Institute of Standards and Technology (NIST)’s integrated colony enumerator (NICE) program [63]. Mammosphere formation was determined by examining the mammosphere formation efficiency (MFE) (%) [64].

### 4.6. Cell Proliferation

Breast cancer cells were cultured at 1 × 10^6^ (MDA-MB-231) and 1.5 × 10^6^ (MCF-7) cells in a culture plate for 1 day and incubated with 6-methoxymellein (0, 0.5, 0.8, 1, 2 and 3 mM) for 1 day. Proliferation was determined by using the EZ-Cytox assay kit (DoGenBio, Seoul, Korea). The optical density at 450 nm (OD_450_) was measured by using a plate reader (VERSA max, Molecular Device, San Jose, CA, USA).

### 4.7. Colony Formation Assay

We used a previously described method for the colony formation assay [36]. Breast cancer cells were cultured in a 6-well plate (1000 cell/well) and cultured with 6-methoxymellein in culture media. After 7 days, the media was removed, washed with 1x PBS, fixed with 3.7% formaldehyde, and stained for 30 min with 0.05% crystal violet. The images were captured by using a scanner.

### 4.8. Flow Cytometric Assay of CD44^+^/CD24^−^ Expression

We followed a previously described method [65]. After treatment of 6-methoxymellein for 24 h, MDA-MB-231 cells were cultured and dissociated into single cells. A total of 1 × 10^6^ cancer cells were labeled with FITC-labeled anti-CD44 and PE- labeled anti-CD24 at 4 °C for 20 min. Then, the cells were washed with 1 × PBS and analyzed using an Accuri C6 flow cytometer (BD, San Jose, CA, USA).

### 4.9. Migration

We followed a previously described method for the migration assay [66]. Migration was examined in 12-well inserts with polycarbonate membranes (Merck, Darmstadt, Germany). MDA-MB-231 and MCF-7 cancer cells were cultured in 200 µL DMEM containing 1% FBS with DMSO or 6-methoxymellein and cultured in the upper chamber (2 × 10^5^ cells/chamber). The bottom chamber was covered with 750 μL of DMEM containing 20% FBS. The cells were incubated for 1 day at 37 °C in an incubator (37 °C, 5% CO_2_). The lower surfaces of the inserts were fixed and stained with 0.03% crystal violet, and images were captured by using a light microscope.

### 4.10. RT-qPCR

We used a previously described method [35]. Total RNA of MDA-MB-231 mammospheres was isolated. RT-quantitative PCR was performed by using an RNA-direct SYBR qPCR kit (Toyobo, Japan). The specific primers for IL-6, IL-8 and β-actin are described in Appendix A.

### 4.11. Western Blotting

We used a previously described method for western blotting [35]. Proteins samples (20 μg/well) of mammospheres treated with 6-methoxymellein were isolated and run on a 10% SDS-PAGE gel. The separated protein gels were transferred to a PVDF membrane (Millipore, Billerica, MA, USA). Membranes were incubated at room temperature for 1 h in Odyssey blocking buffer in PBST (0.1% Tween 20). The membranes were incubated at 4 °C overnight with the following primary antibodies: NF-κB p65, LF-MA30327, NF-κB p50, sc-8414; Oct4, LF-MA30482 (AbFrontier, Seoul, Korea); c-Myc (Cell Signaling Technology, Denver, CO, USA); Sox-2, Lamin B and β-actin (Santa Cruz Biotechnology, Dallas, TX, USA). After the membranes were washed with 1x PBST, they were incubated with IRDye 800CW and 680RD-conjugated secondary antibodies for 1 h, and band densities were examined by using an ODYSSEY CLx (LI-COR, Lincoln, NE, USA).

### 4.12. EMSA

Nuclear proteins were prepared as described previously method [67]. EMSA experiments for NF-κB binding were performed using an IRDye 700-labeled NF-κB oligonucleotide. Samples containing the NF-κB/700 dye oligo complex were run on a native 4% PAGE gel, and the fluorescence of the PAGE gel was scanned and captured by using an ODYSSEY CLx machine (LI-COR).

### 4.13. Quantification of Extracellular Human IL-6 and IL-8 Cytokines Using the Cytometric Bead Array (CBA) Human Inflammatory Cytokine Assay Kit

We followed a previously described method [36]. MDA-MB-231 mammospheres were incubated for 5 days and then cultured with 6-methoxymellein for 2 days. The extracellular IL-6 and IL-8 concentrations were analyzed using the BD CBA Human Inflammatory Cytokine Assay Kit. We followed the manufacturer’s protocol (BD, San Jose, CA, USA). Fifty microliters of bead assay buffer, equal volumes of cultured media and 50 μL of PE-labeled capture antibody solution were mixed into each tube. The sample mixes were incubated at room temperature protected from light for 3 h, washed with 1 × washing buffer. The pellet was suspended in reaction buffer and analyzed by flow cytometry Accuri C6 (BD, San Jose, CA, USA).

### 4.14. Statistical Analysis

Our results are presented as the mean *±* standard deviation (SD). Our results were analyzed using Student’s *t*-test. A *p*-value less than 0.05 was considered statistically significant (GraphPad Prism 7 Software).

## 5. Conclusions

6-Methoxymellein from carrots identified by mass spectrometry and NMR, acts as a mammosphere formation inhibitor. 6-Methoxymellein inhibits the proliferation, migration, and colony and mammosphere formation of breast cancer cells and decreases the subpopulation of CD44^+^/CD24^−^ and the expression of c-Myc, Sox-2 and Oct4 proteins. In addition, the compound reduces nuclear NF-κB p65 and p50 protein expression, subsequently decreasing the transcript expression and secretion of IL-6 and IL-8 by mammospheres. Our data suggest that 6-methoxymellein suppresses the NF-κB signaling pathway and reduces the expression of c-Myc, Sox-2 and Oct4, may be an inhibitory compound against BCSCs.

## Figures and Tables

**Figure 1 molecules-25-04374-f001:**
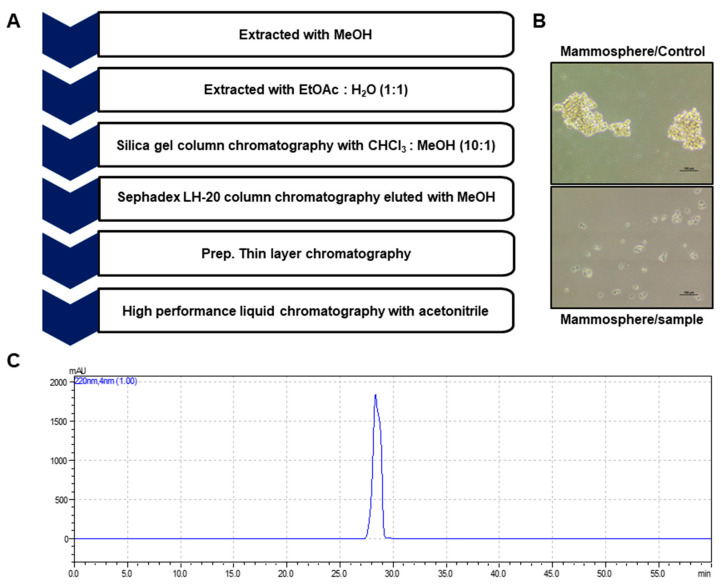
Isolation of breast CSC inhibitors from carrot extracts based on mammosphere formation assays. (**A**) The purification procedure for the mammosphere inhibitor. (**B**) Inhibitory effect of carrot extracts on mammosphere growth. The mammospheres were incubated with carrot extracts or DMSO. The mammospheres were photographed with a microscope at 10× magnification (scale bar = 100 µm). (**C**) Purified samples of carrot extracts were analyzed by HPLC chromatogram.

**Figure 2 molecules-25-04374-f002:**
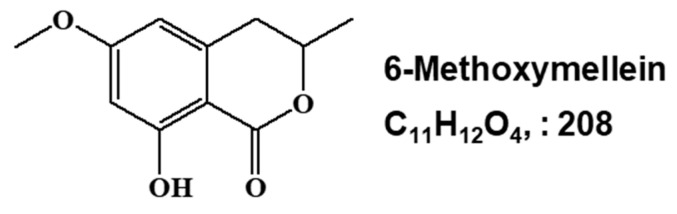
Chemical structure of the CSC inhibitor isolated from carrots. Chemical structure of 6-methoxymellein.

**Figure 3 molecules-25-04374-f003:**
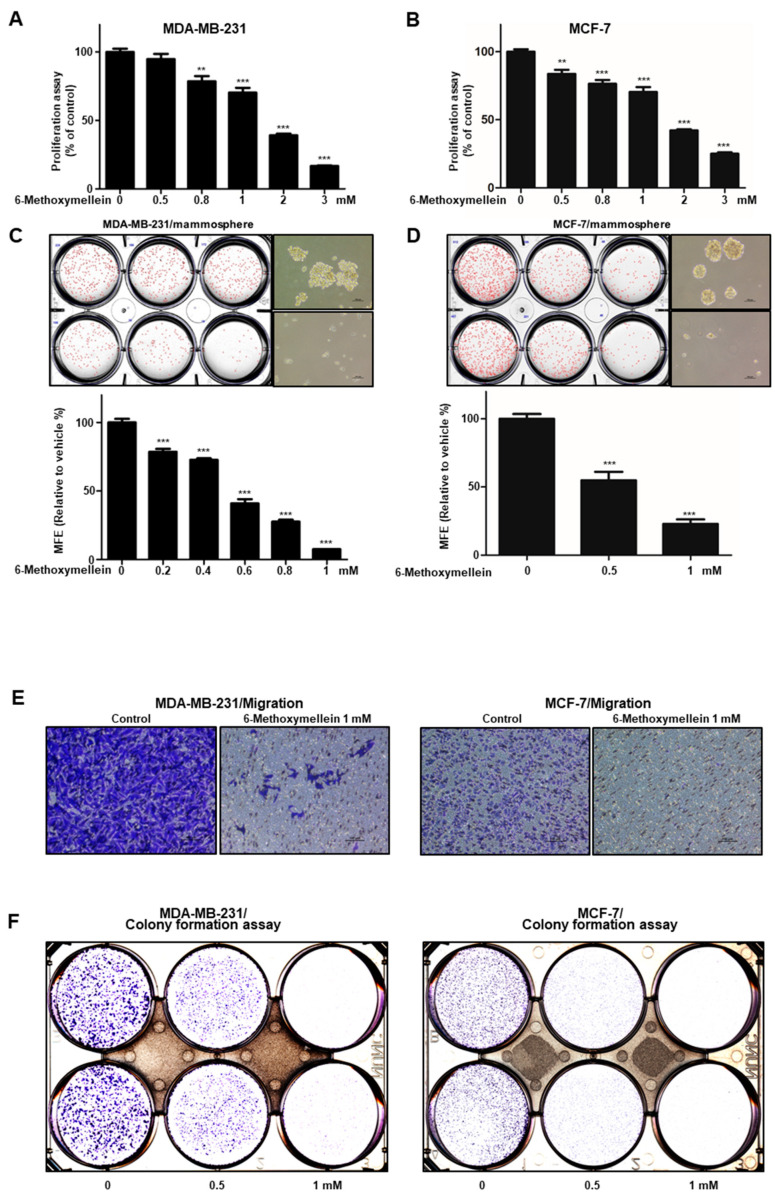
The antiproliferation and mammosphere formation inhibitory effect of 6-methoxymellein. (**A**,**B**) The antiproliferation ability of 6-methoxymellein was assayed with an EZ-Cytox assay kit using breast cancer cells treated with 6-methoxymellein. (**C**,**D**) To establish the inhibitory effect of 6-methoxymellein on mammosphere formation, breast cancer cells were cultured in 6-well plates (ultralow attachment) with CSC culture media containing increasing concentrations of 6-methoxymellein or DMSO alone after 7 days. The images were captured and are representative mammospheres by microscopy at 10× magnification (scale bar = 100 µm). (**E**,**F**) The migration of MDA-MB-231 and MCF-7 cells was determined by Transwell assay after exposure to 6-methoxymellein (scale bar: 100 μm). 6-Methoxymellein inhibits the colony formation of MDA-MB-231 and MCF-7 cells. Our representative data were collected. Our data are represented as the mean ± SD; ** *p* < 0.01; *** *p* < 0.001.

**Figure 4 molecules-25-04374-f004:**
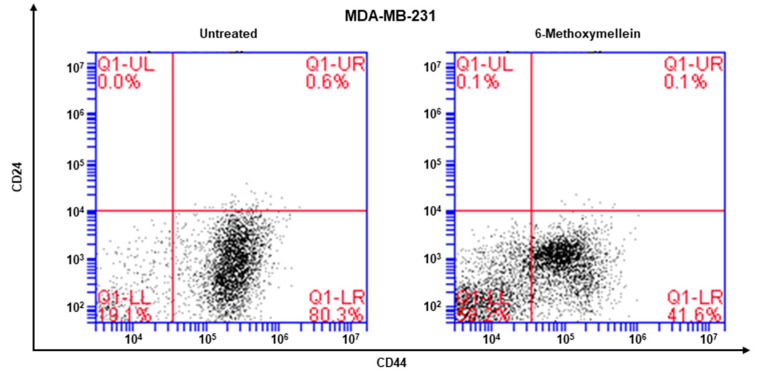
6-Methoxymellein reduces the proportion of CD44^+^/CD24^−^ cells. MDA-MB-231 cells were treated with 6-methoxymellein (1 mM) for 24 h. The CD44^+^/CD24^−^ cell proportion was assessed by an Accuri C6 flow cytometer. The red cross was used for binding of a control antibody.

**Figure 5 molecules-25-04374-f005:**
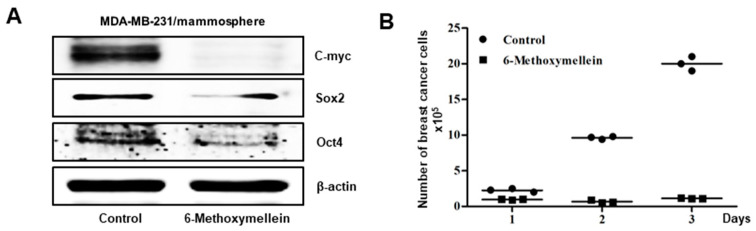
6-Methoxymellein suppresses the protein expression of CSC markers and inhibits mammosphere. (**A**) Westernblot analysis of the c-Myc, Sox-2 and Oct4 proteins of mammospheres treated with 6-methoxymellein for 2 days. (**B**) 6-Methoxymellein inhibits mammosphere growth. Mammospheres treated with 6-methoxymellein were split into single cells and plated in 6-cm culture plates. 1, 2 and 3 days later, the cells were quantified.

**Figure 6 molecules-25-04374-f006:**
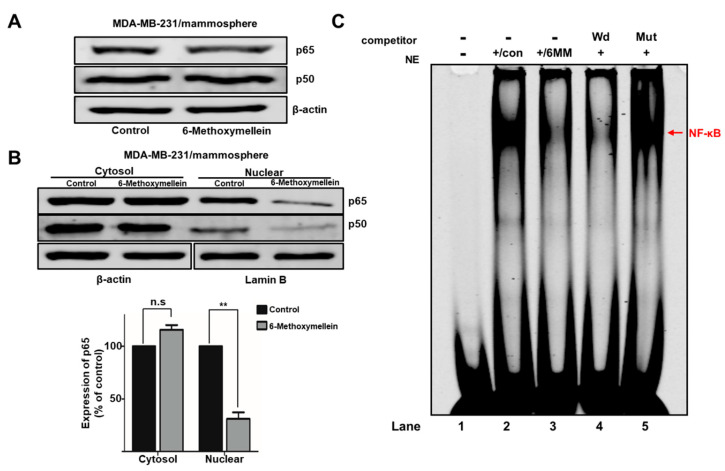
The effect of 6-methoxymellein on the NF-κB signaling pathway. (**A**) The total protein level of NF-κB p65 and NF-κB p50 were assayed in mammospheres under 6-methoxymellein for 48 h using immunoblot analysis. (**B**) The levels of nuclear NF-κB p65 and NF-κB p50 were assayed in mammospheres under 6-methoxymellein (1 mM) or DMSO. 6-Methoxymellein blocks the translocation of NF-κB p65 and NF-κB p50 in mammospheres. (**C**) EMSAs of nuclear protein in MDA-MB-231 cell-derived mammospheres under 6-methoxymellein. The nuclear extracts were incubated with the NF-κB probe and run by 4% native PAGE. Lane 1: IR700-NF-κB probe only; lane 2: nuclear extracts with the IR700-NF-κB probe; lane 3: 6-methoxymellein-treated nuclear proteins with the IR700-NF-κB probe; lane 4: untreated nuclear proteins incubated with the self-competitor (100×) oligo; lane 5: untreated nuclear extracts incubated with the mutated-NF-κB (100×) oligo. The arrow shows the DNA/NF-κB protein complex from nuclear lysates of mammospheres. Our data are presented as the mean ± SD of three independent experiments. ** *p* < 0.05 versus the control group.

**Figure 7 molecules-25-04374-f007:**
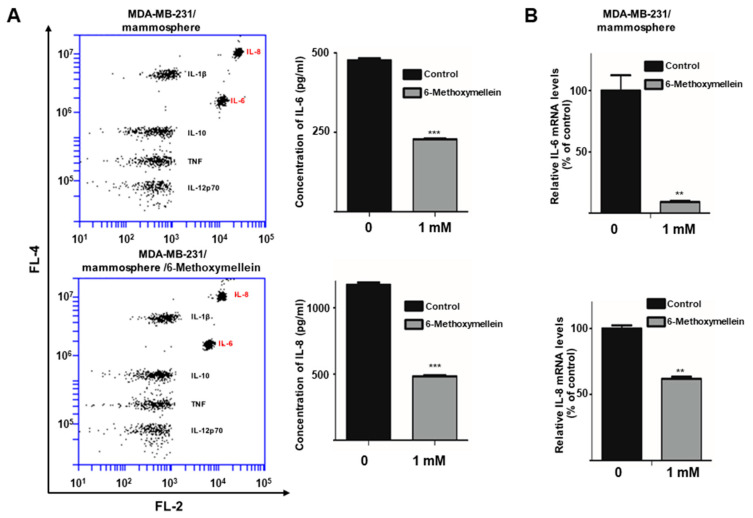
6-Methoxymellein suppresses the secretion and transcription of IL-6 and IL-8. (**A**) The CBA Human Inflammatory Cytokine Assay Kit was used to assay the secretion of cytokines in mammosphere culture media treated with 6-methoxymellein or DMSO. (**B**) Transcript levels of the IL-6 and IL-8 genes were assessed in 6-methoxymellein-treated mammospheres using specific RT-qPCR primers. Our data are represented as the mean ± SD. ** *p* < 0.05 and *** *p* < 0.01 versus the control group.

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
