# Peer review of "6-Methoxymellein Isolated from Carrot (Daucus carota L.) Targets Breast Cancer Stem Cells by Regulating NF-κB Signaling"

_molecules, 2020, doi:10.3390/molecules25194374_

Round 1

Reviewer 1 Report

The manuscript present the evaluation of the 6-Methoxymellein on the the proliferation, migration and mammosphere formation of breast cancer cells. Moreover the impact on the subpopulation of
CD44+/CD24- 336 and the expression of c-Myc, Sox2 and Oct4 proteins. In addition, the Authors demonstrated the effect on the NF-kBp65. 

Comments:

The Authors should explain  in Introduction or Discussion section using in the study these protein :-Myc, Oct4 and Sox2

The Authors should present the effect on the p50 subunits of NF-kB. 

In my opinion it would be advisable designation expression of c-Myc, Oct4, Sox-2 and NF-kBp50 and p65 subunits 

Author Response

We submit our reply to the review1 report.

Reviewer 2 Report

The manuscript by Liu et al., describes that 6-methoxymellein isolated from carrot (Daucus 2 carota L.) targets breast cancer stem cells by regulating NF-κB signaling. The specific concerns are as follows:

  1. Although the authors showed that 6-methoxymellein inhibited NF-κB nuclear shuttling, the ratio of increase of cytosolic fraction did not fit decrease of nuclear fraction.
  2. In Figure 6C, the data interpretation is conflicting.
  3. The authors did not show the direct evidence of NF-κB involvement in this regulation.

Author Response

We submit our reply to review 2 report.

Round 2

Reviewer 1 Report

I accept in this form 

Reviewer 2 Report

The authors have addressed the concerns I raised.